

# Comparative genomic analysis of emerging non-typeable *Haemophilus influenzae* (NTHi) causing emerging septic arthritis in Atlanta

Brianna J. Bixler[1], Charlotte J. Royer[1], Robert A. Petit III[1], Abraham G. Moller[1], Samantha Sefton[1,2], Stepy Thomas[1,2], Amy Tunali[1,2], Lauren F. Collins[1,3], Monica M. Farley[1], Sarah W. Satola[1] and Timothy D. Read[1]

[1] Division of Infectious Diseases, Department of Medicine, Emory University, Atlanta, GA, United States
[2] Georgia Emerging Infections Program, Atlanta, GA, United States
[3] Ponce de Leon Center, Grady Health System, Atlanta, GA, United States

Corresponding author
Timothy D. Read, tread@emory.edu

## ABSTRACT

**Background:** *Haemophilus influenzae* is a Gram-negative bacterium that can exist as a commensal organism or cause a range of diseases, from ear infections to invasive conditions like meningitis. While encapsulated *H. influenzae* strains have historically been linked to severe diseases, non-typeable *Haemophilus influenzae* (NTHi) strains, lacking an intact capsule locus, have emerged as the leading cause of invasive *H. influenzae* infections, particularly following the widespread use of the *H. influenzae* serotype b (Hib) vaccine.

**Methods:** In response to a significant increase in invasive NTHi infections among persons living with HIV in metropolitan Atlanta during 2017–2018, we conducted a comparative genomic analysis of two predominant NTHi clones, C1 and C2, identified during this period. These clones correspond to multilocus sequence types ST164 and ST1714, respectively. We analyzed the genomic characteristics of C1 and C2 using whole genome sequencing data and compared them to a broader pangenome of *H. influenzae* strains to identify potential virulence factors and genetic adaptations.

**Results:** Both C1 and C2 isolates were highly related within their clusters, with C1 showing a maximum of 132 SNPs and C2 showing 149 SNPs within their respective core genomes. Genomic analysis revealed significant deletions in known virulence genes, surprisingly suggesting possible attenuation of virulence. No unique accessory genes were identified that distinguished C1 and C2 from other *H. influenzae* strains, although both clusters exhibited a consistent loss of the *pxpB* gene (encoding 5-oxoprolinase subunit), replaced by a mobile cassette containing genes potentially involved in sugar metabolism. All C1 and C2 isolates showed potential enrichment in accessory genes associated with systemic infections.

**Conclusions:** Our study suggests that while C1 and C2 clones possess some genetic markers potentially linked to systemic infections, there are no definitive unique genetic factors that distinguish these clones as more virulent than other *H. influenzae* strains. The expansion of these clones in a vulnerable population may reflect both chance introduction and potential adaptations to the host environment. Further

research is needed to understand the implications of these genetic findings on the clinical management and prevention of invasive NTHi infections.

## INTRODUCTION

*Haemophilus influenzae* is a Gram-negative bacterium that can live on human mucosal surfaces without causing an infection but can also be associated with ear and respiratory infections or more invasive diseases, such as bacteremia or meningitis (*Butler & Myers, 2018*; *Dworkin, Park & Borchardt, 2007*; *Foxwell, Kyd & Cripps, 1998*; *Xiao et al., 2023*). *H. influenzae* strains expressing polysaccharide capsule genes a-f, containing capsule genes on the IS1060 transposon, have historically been associated with more serious invasive disease (*Satola, Napier & Farley, 2008*; *Satola et al., 2007*; *Kroll, Loynds & Moxon, 1991*). Since the routine use of the *H. influenzae* serotype b (Hib) vaccine in the 1990s, strains lacking the intact capsule locus (NTHi; non-typeable *Haemophilus influenzae*) have replaced encapsulated strains as the leading cause of invasive *H. influenzae* disease (*Eskola et al., 1990*; *Soeters et al., 2018*).

In recent years, a CDC-funded active population-based surveillance program was leveraged to evaluate a sharp increase in the rate of NTHi infection among persons living with HIV in 2017–2018 compared to prior years evaluated 2008–2016 and identified that the cases primarily occurred in Black men who have sex with men that had a high prevalence of septic arthritis (*Collins et al., 2019*). Pulsed-field gel electrophoresis typing among NTHi cases aged 18–55 years identified two expanded NTHi clones, named clusters 1 and 2 ("C1" and "C2") as predominant in the 2017–2018 NTHi cases. Whole genome shotgun analysis identified C1 and C2 isolates as corresponding to multilocus sequence types ST164 and ST1714, respectively. None of the C1/C2 strains contained capsule genes but all C1 contained the IS1016 transposon gene. Additionally, in the C1 isolates, there were two genes flanking the IS1016 genes (aquaporin family protein (*Fleischmann et al., 1995*) and thiamine ABC transporter substrate–binding subunit (*How et al., 2025*)), that are homologs of genes at this locus in encapsulated strains (*Fleischmann et al., 1995*; *Rodríguez-Arce et al., 2019*). This suggested that the site between the two flanking genes was a hotspot for IS1016 mediated insertion. Although ST164 and ST1714 were close relatives within the *H. influenzae* species phylogeny, their last common ancestor clearly predated the likely timing of the Atlanta cluster of infections, suggesting that two independent clusters of infections were occurring concurrently. Geospatial analysis of NTHi cases in metropolitan Atlanta revealed temporal-geographic separation between cases by cluster type as C1 and C2 and further, significant temporal-geographic aggregation of C1 cases from January–December 2017 in a certain geography compared with C2 cases (*Collins et al., 2019*).

The NTHi C1 and C2 isolates may have a greater genetic propensity to cause more serious invasive disease (*i.e.*, septic arthritis) or alternatively, their expansion reflected chance introduction into a vulnerable population and transmission within social networks. In this study, we performed a comparative genomic analysis of the NTHi C1 and C2 isolates originally identified in metropolitan Atlanta in the context of the larger pangenome of *H. influenzae* strains and other *Haemophilus* species to identify potential features that may suggest enhanced virulence in the cluster strains.

## MATERIALS AND METHODS

### Oxford nanopore sequencing and hybrid assembly with Illumina data

One *Haemophilus influenzae* isolate was randomly chosen from each of the C1 and C2 clusters for hybrid assembly of Oxford Nanopore minIOn and Illumina data to produce complete reference sequences (referred to as C1-1 and C2-1). Genomic DNA was extracted using the Promega Wizard Genomic DNA Purification Kit. Sequencing libraries prepared using the SQK-LSK109 1D ligation sequencing kit and sequenced on a FLO-FLG001 Flongle flow cell, yielding 496.9 Mb and 552.6 Mb of raw reads (~267x and ~297x coverage) for C1-1 and C2-1, respectively. The *H. influenzae* C1-1 and C2-1 genomes were then assembled from Nanopore and Illumina paired-end reads using Unicycler (*Wick et al., 2017*). Assemblies were deposited in the NCBI assembly database under GCF_044096575 (C1-1) and GCF_044096445 (C2-1). Raw data was deposited to NCBI as part of accession PRJNA544724.

### *H. influenzae* virulence protein database

We created a non-redundant BLAST database of all of the *H. influenzae* virulence factors defined in the Virulence Factor Database (http://www.mgc.ac.cn/VFs/main.htm) and the Victors database (https://phidias.us/victors/) (*Sayers et al., 2019*; *Chen, 2004*). We also manually curated the recent literature to identify genes and sequences associated with virulence that may not have been included in the VFDB or Victors databases. In April 2024 we performed a search in pubmed for articles containing the term "'*Haemophilus influenzae*" [All Fields] AND ("pathogenicity" [MeSH Subheading] OR "pathogenicity" [All Fields] OR "virulence" [All Fields] OR "virulence" [MeSH Terms] OR "virulences" [All Fields] OR "virulent" [All Fields]) 'published between 2004–2024. Gene names associated with virulence were identified from reading the articles and were matched to uniprot accessions from *H. influenzae* or related gram-negative bacteria. From this search, we added gene and protein sequences of 105 *H. influenzae* reference genes described in *Pinto et al. (2019)*.

We created a BLAST database of all the protein sequences in our table. To match a query protein against our database we used blastp v2.10.1+ with default parameters and used a threshold of 80% identity to define a hit.

### Processing whole genome data

We used "bactopia search" from the Bactopia software package (*Petit & Read, 2020*) to identify 4,842 publicly available *H. influenzae* genomes from the Sequence Read Archive

*database*. Whole genome shotgun Illumina paired-end fastq data files were downloaded from NCBI SRA database on 05-April-2021 and processed using Bactopia (v1.6.0) (*Petit & Read, 2020*). The multi-locus sequence type (MLST) schema for *H. influenzae* from PubMLST.org (*Jolley, Bray & Maiden, 2018*) was included and completed genomes for C1-1 and C2-1 were added as optional datasets.

Within the Bactopia pipeline the reads were cleaned and error-corrected using BBDuk (v38.86, (*Bushnell, 2014*)) and Lighter (v1.1.2, (*Song, Florea & Langmead, 2014*)). Processed reads were assembled with SKESA (v2.4.0, (*Souvorov, Agarwala & Lipman, 2018*)) using Shovill (v1.1.0, (*Seemann, 2014*)). Assembly quality metrics were determined with assembly-scan (v0.3.0 (*Petit, 2017*) and CheckM (v1.1.3, (*Parks et al., 2015*)). The species composition of the assembly was determined by screening against a minmer sketch of GenBank (*Clark et al., 2016*) using sourmash (v3.5.0, (*Titus Brown & Irber, 2016*)). The MLST was determined using BLAST+ (V2.10.1, (*Camacho et al., 2009*)) and Ariba (v2.14.6, (*Hunt et al., 2017*)).

## Generating a representative dataset

Due to the large number of samples collected and diverse sequence types among them, we decided to create a smaller representative dataset for downstream analyses. This set would include at least one high quality sample from each sequence type, as well as all high-quality Georgia-specific samples. Both capsulated and unencapsulated samples were included in this set to enable a broader pan-genome comparison when assessing unique traits of C1 and C2. First, we generated an intermediate group of high-quality samples that would include candidates for the final representative dataset. We used the "summary" tool from Bactopia to aggregate the quality control results for all genomes into a single table. This included assignment of a rank, either gold, silver, or bronze, based on coverage, mean per-read quality, mean read length, and total number of contigs. The cutoffs for each category were:

Gold:
Coverage >= 100x
Quality >= Q30
Read Length >= 95 bp
Total Contigs < 100
Silver:
Coverage >= 50x
Quality >= Q20
Read Length >= 75 bp
Total Contigs < 200
Bronze:
Coverage >= 20x
Quality >= Q12
Read Length >= 49 bp
Total Contigs < 500

Any samples lower than a bronze quality rank were dropped. The average nucleotide identity (ANI) between C1 and all other genomes was calculated with FastANI (*Jain et al., 2018*) (v1.32 (*Souvorov, Agarwala & Lipman, 2018*)) to determine relatedness between samples.

To generate the intermediate data from the post-QC samples, a custom Python script ('generate-representative-set.py') was created. The script tags representative samples by prioritizing Georgia samples first, then samples with ANI greater than a specified cutoff. For any remaining non-Georgia non-C1/C2 samples, it selects representatives based on the sample rank (gold, silver, bronze), CheckM completeness, and the number of contigs, prioritizing higher completeness and fewer contigs. Samples are excluded if they do not meet the criteria for paired-end reads, fail quality checks, or have undesirable sequence types, and the exclusion reasons are documented in an output file. The script updates exclude and representative files accordingly, utilizing Python library argparse for command-line argument parsing, os for file operations, and collections for ordered dictionary management.

For the final representative dataset, we further reduced the intermediate data to the best single representatives of each sequence type. Any genomes with more than 500 assembled contigs, those that did not have a sequence type determined, or were not screened as "*Haemophilus influenzae*" by sourmash (*Titus Brown & Irber, 2016*), were excluded. The representative set included all high-quality C1, C2, and non-cluster isolates from the GA EIP investigation from this study and all genomes that had an ANI greater than ANI of the most distant C2 genome from C1-1 (a member of C1). This was done to ensure that no highly divergent samples were included. If an ST was identified, but not yet included, a genome was picked prioritizing its quality rank, assembly completeness, and total number of contigs. The final representative set was used to determine the pan-genome using PIRATE (v1.0.4, (*Bayliss et al., 2019*)). A recombination masked core genome alignment was produced using ClonalFrameML (*Didelot & Wilson, 2015*) using a maximum-likelihood calculation. A phylogenetic tree based on this alignment was created with IQ-TREE (v2.0.3, (*Nguyen et al., 2015*; *Hoang et al., 2018*)) with default parameters.

## Intra-clade comparisons

SNPs were called using the bactopia-tools workflow for snippy (*Seemann, 2014*). To understand if there was any variability across the entire genomes of C1 and C2 samples, reads from each sample were individually mapped to the completed reference genome of each and potentially deleted regions were assessed by lower-than-expected coverage in regions. In 1,000 bp windows, sliding every 250 bp, reads were counted, adjusted for reads per kilobase per million (RPKM), and selected if one of the samples had 0 reads over that bin. Adjacent regions of zero coverage were combined into a single region of interest and coverage was converted into a binary presence or absence.

## Genome-wide association study (GWAS)

We used the pangenome GWAS tool Scoary (*Brynildsrud et al., 2016*) to investigate whether blood or sputum-associated genes were enriched in C1s and C2s. We randomly

choose one representative sample from each ST collected from either blood or sputum to reduce bias introduced by oversampling a small number of STs with many genomes. This resulted in a set of 146 blood and 87 sputum-associated genomes that were used in the GWAS.

### Recombination analysis

The core-genome alignment of 50 randomly chosen STs (including C1 and C2) with the *sample()* function in R with set.seed = 9,696 and recombination events were predicted by ClonalFrameML. The output includes a recombination-free alignment and a set of predicted recombination events, which were subsequently analyzed to understand the patterns and impact of horizontal gene transfer on the core-genome evolution of the selected STs

### AMR analysis

We used the AMRFinder tool (*Feldgarden et al., 2021*) with default settings to identify antimicrobial resistance (AMR) genes in the analyzed sequences. The presence of these AMR genes was then plotted onto the phylogenetic tree to visualize their distribution across different strains, highlighting potential evolutionary relationships and patterns of resistance.

### Inter-species comparative genomic analyses

The MAUVE (*Darling et al., 2004*) tool was used for comparative analyses between NTHi and other *Haemophilus* reference species to identify shared or unique ancestral sequences. The reference species *H. aegyptius* (GCA_900475885), *H. parainfluenzae* (GCA_000191405), *H. haemolyticus* (GCA_900477945) and *H. seminalis* (GCA_008605885) were compared to our NTHi samples using MAUVE.

### Data availability

Commands, code, and data used in this study (including the "generate-representative-set.py" script, PIRATE gene family results, and curated blastp virulence gene database) are available at https://doi.org/10.5281/zenodo.14861587 and https://doi.org/10.5281/zenodo.14861572.

## RESULTS

### Genetic variation within the C1 and C2 clusters

There were 26 C1 and 23 C2 isolates with whole genome shotgun (WGS) Illumina data from the original Atlanta investigation from 2008–2018. One isolate was picked at random from each group ("C1-1" and "C2-1") for closing using Oxford nanopore sequencing techolony. The final C1-1 hybrid assembly included one circular 1.875 Mb contig, while C2-1 included one circular 1.885 Mb and one circular 37.6 Kb plasmid. Table S1 shows hits of the virulence gene database to coordinates on the assembled C1-1 and C2-1 genomes.

The C1 isolates were found to be highly related, with a maximum distance of 132 SNPs in the 1.55 Mb core genome alignment (Fig. 1A). All C1 isolates were of the same sequence type, ST164. There were 29 unique regions of low coverage found in at least one sample

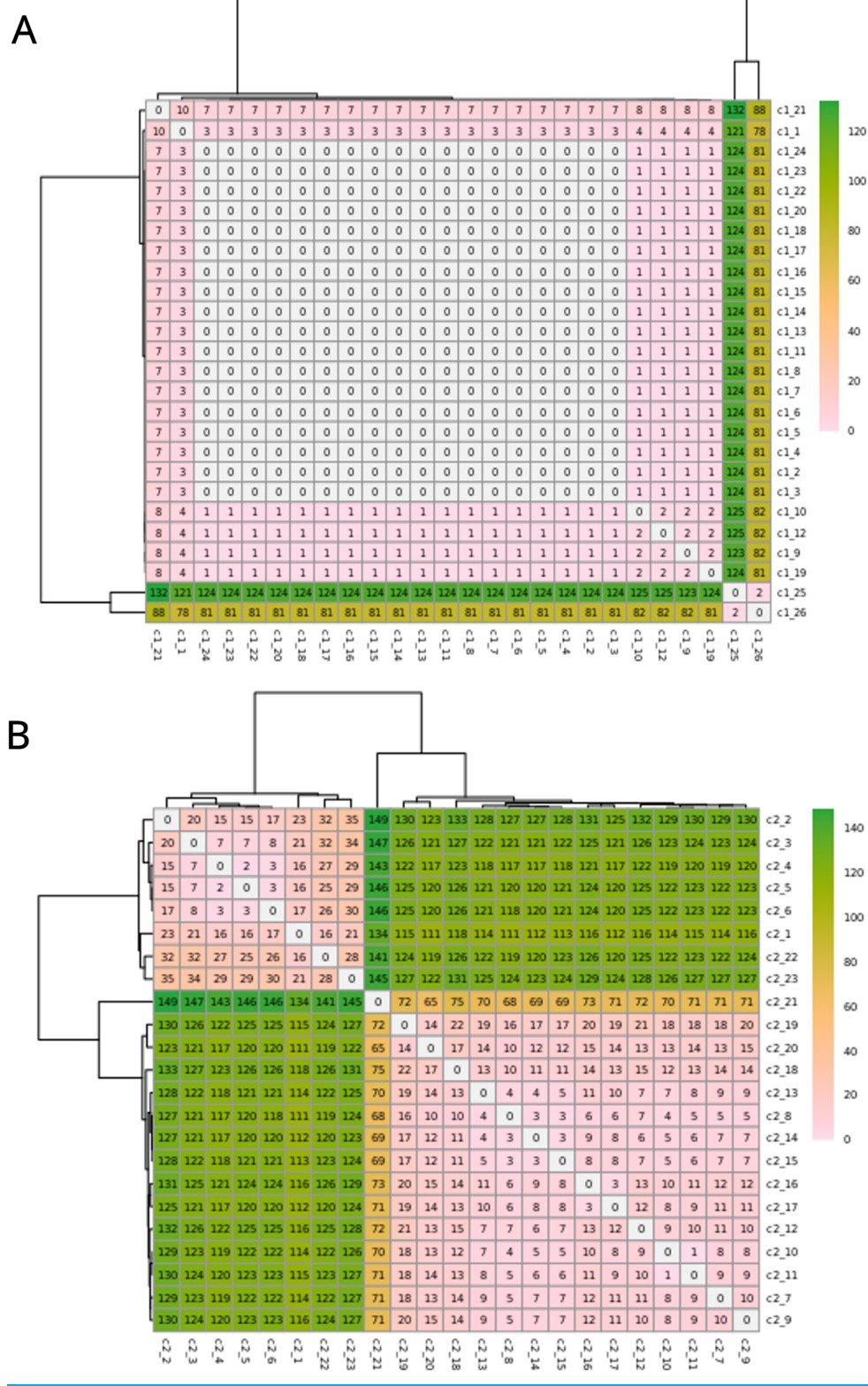

**Figure 1 SNPs in the core alignments of C1 and C2.** SNP distance matrices of core alignments of all C1-C1 comparisons (A) and all C2-C2 comparisons (B). Columns and rows were clustered using hierarchical clustering with the complete linkage method (default in R's pheatmap function).

(hereafter "deletion regions") in the C1 isolates compared to the reference genome, ranging from 1 to 17 kb and encompassing 48 genes (Fig. 2A). Of these deletion regions, 15 genes had a significant match to *H. influenzae* virulence genes (*ghfB, ghfC, ghfD, ghfE, hafB, hafC, hafD, hafE, hgpB, hgpD, hifB, hifC, hifD, hifE*, and *tbpB)*. The C2 isolates were also found to be highly related, all representing the same sequence type, ST1714, and having a maximum distance of 149 SNPs in the 1.60 Mb core genome alignment (Fig. 1B). The C2 cluster was separated into 2 subclades, with sample SNP distances as low as 1 and as high as 35 SNPs within these sub-groups. In a similar manner to C1 analysis, there were 13 large deletion regions (~1 to 32 kb), spanning 24 annotated genes (Fig. 2B), 6 of these genes matched suspected virulence genes: (*hgpB, hgpD, licA, licB, licC, mrsA/glmM)*. Of the genes that had annotated gene names, five were deleted in at least one sample in both C1 and C2 (*eamA, ninG,* sRNA-Xcc1, sRNA-Xcc1, and *tolB*). None of these genes are found in the virulence factor database or the literature-curated database.

## Few accessory genes distinguish the C1 and C2 clusters from other *H. influenzae* strains

Our public data search yielded 536 distinct MLST sequence types from a total of 4,842 available *H. influenzae* genomes. From this we randomly selected 536 ST representatives (one for each ST), in addition to 26 C1 isolates, and 23 C2 isolates for the pangenome analysis. This final group of 583 samples is referred to as the "representative dataset". The pangenome consisted of 6,560 gene families, of which 1,368 were core (>= 95% genomes), 1,107 intermediate accessory (5% < x < 95%) and 4,085 rare accessory (x = <5%) (Fig. S1). Overall, we found C1 and C2 were part of a closely related subclade of NTHi strains (Fig. 3).

We found that there were no accessory genes unique to either C1 or C2, or both C1/C2 (*i.e.*, present in one or both clades and not found in other MLSTs). There were also no core genes missing in only C1 or C2, or both. While there were no genes absent in C1 and/or C2 that were present in 100% of the rest of the *H. influenzae* pangenome, there was one gene family, identified as *pxpB*, that was lost in 100% of the C1 and C2 isolates and present in more the 90% of rest of the population of strains (Fig. 3). There were 20 gene families that were 'rare' in the context of the *H. influenzae* pangenome (*i.e.*, found in < 10% STs) present in C1 but not C2 isolates (Table 1). Further, there were seven 'rare' gene families unique to C2, that were not identified in C1 strains (Table 2). There were no rare genes shared by both C1 and C2.

We used blastp to investigate the presence or absence of virulence factors in the predicted proteins from each of the 583 representative samples. We found the number of sequences present was comparable across most STs, with a total of 167 hits across all samples out of 207 total tested sequences (Fig. S2A). When comparing C1 and C2, there were two sequences unique to C1, three unique to C2, and 129 common between them (Fig. S2B). None of these rare unique genes were in our set of known virulence genes (Fig. S3). There were no known antimicrobial resistance (AMR) genes in the C1, C2 or closely related STs (Fig. S4).
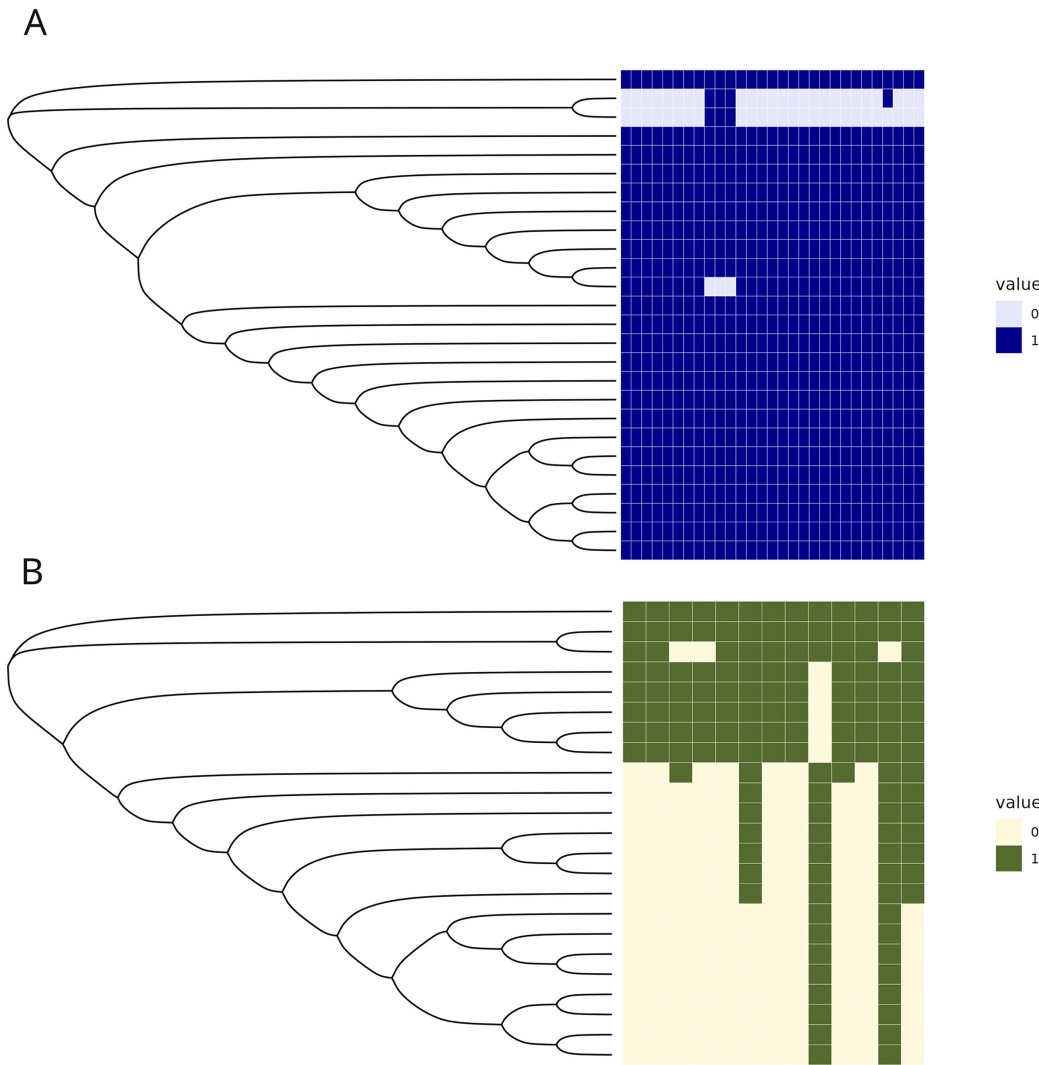

**Figure 2 Regions of low coverage within C1 and C2.** Regions of low coverage within clusters C1 (A) and C2 (B). Regions were defined by counting reads over 1,000 bp windows, sliding every 250 bp. Reads were counted, manually converted to rpkm, and selected if one of the samples had 0 reads over that bin. Adjacent regions of zero coverage were combined into a single area of interest and coverage was converted into a binary presence or absence. Areas of low coverage were plotted for each C1 and C2 sample, where each column represents a unique region of low coverage found in at least one sample. These regions were plotted in the heatmap below, where 1 represents the presence of a deletion and 0 is the absence of that deletion.

## C1 and C2 isolates possess a unique ancestral mobile cassette

While examining the *H. influenzae* pangenome data, we observed an interesting pattern of gains and losses that correspond to a previously undescribed mobile cassette inserted in the same ancestral region of C1, C2, and a small subset of other sequence types. There were eight genes identified as present in both C1 and C2 but in less than 90% of the rest of the pangenome (Fig. 3). All eight gene families had significant homoplasy, with a consistency index less than or equal to 0.2 on the species core genome tree (Table S1). Within this subset of rare genes, we identified five genes that were likely acquired as a cassette in the

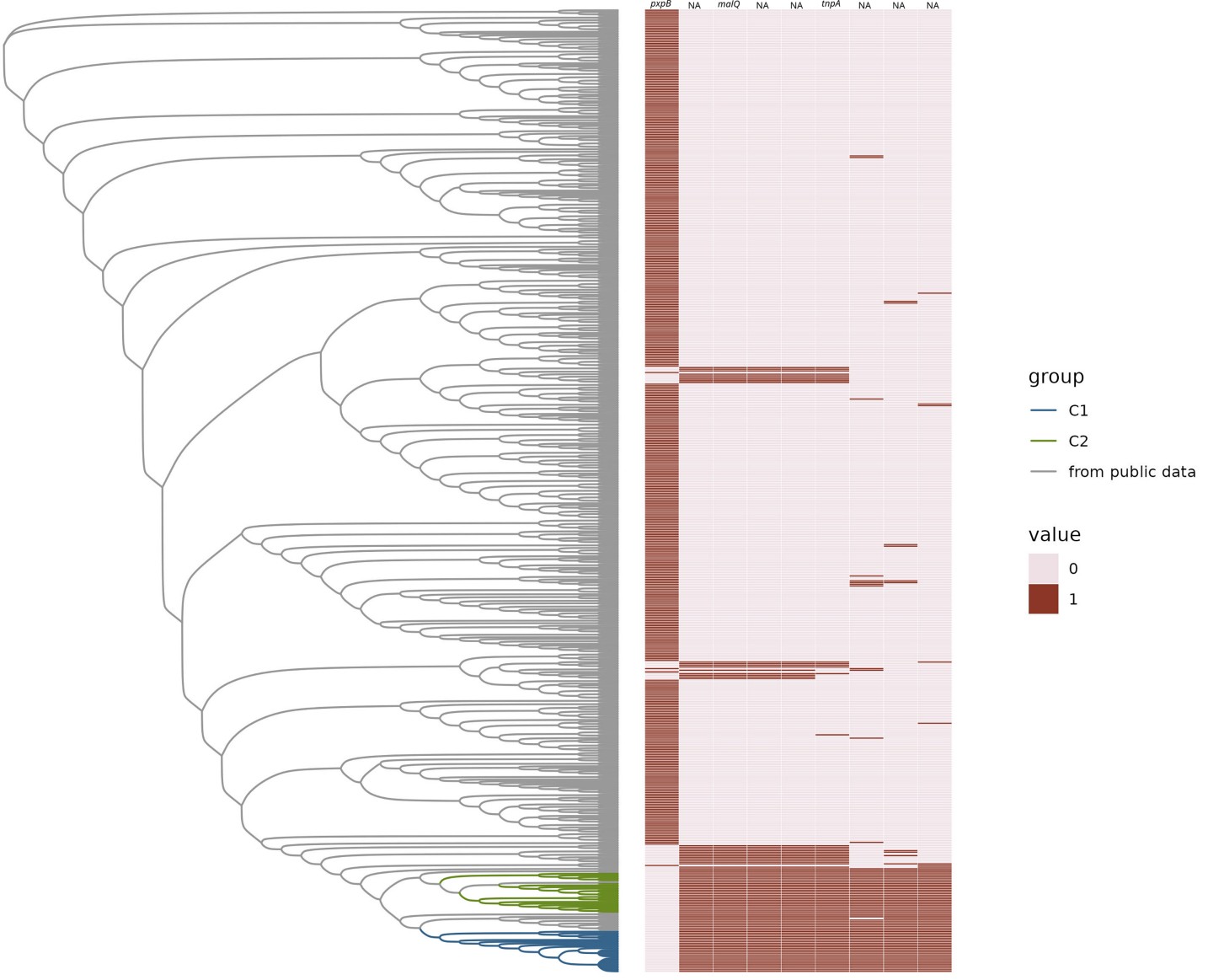

**Figure 3 Gained and lost gene families.** Neighbor-joining tree of representative *H. influenzae* genomes from the public data plus representatives of C1 and C2 subclades represented as a phylogram. The heatmap represents the presence (1) or absence (0) of gene families gained and lost from C1 and/or C2. Gene names are provided at the top of each column.

ancestor and inserted at the ancestral site of the one lost gene family. The gene that was the apparent site for cassette integration in C1 and C2 encoded a protein annotated as 5-oxoprolinase subunit PxpB (*Niehaus et al., 2017*). Using comparative genomic analysis with the MAUVE tool, we found that the five inserted genes were on a cassette of 9,444 bp in C1 and C2. In the same region of the outgroup that lacked the cassette (GCF_014701215.1_ASM1470121v1), there was an intact *pxpABC* gene cluster. When that region is compared to *H. aegyptius*, *H. parainfluenzae*, *H. haemolyticus* and *H. seminalis* using MAUVE, this region was not like any sequence in these reference pathogens.

**Table 1 C1 unique genes.** Genes unique to C1 from pangenome analysis.

| Consensus product | Locus | Length | Percentage of pangenome | C1 | C2 | Not C1C2 |
|---|---|---|---|---|---|---|
| Hypothetical protein | 310,199–309,633 | 567 | 0.10291595 | 26 | 0 | 34 |
| Hypothetical protein | 895,450–895,608 | 159 | 0.09605489 | 25 | 0 | 31 |
| Hypothetical protein | 893,172–892,912 | 261 | 0.12178388 | 25 | 0 | 46 |
| Helix-turn-helix domain-containing protein | 894,885--894,646 | 240 | 0.10977702 | 25 | 0 | 39 |
| Type 11 toxin-antitoxin system RelE/ParE family toxin | 896,804–897,103 | 300 | 0.10120069 | 25 | 0 | 34 |
| Hypothetical protein | 896,581–896,811 | 231 | 0.10120069 | 25 | 0 | 34 |
| Hypothetical protein | 33,521–34,678 | 1,185 | 0.06174957 | 26 | 0 | 10 |
| WYL domain-containing protein | 32,591–33,403 | 813 | 0.07032590 | 26 | 0 | 15 |
| Hypothetical protein | 896,104–896,337 | 234 | 0.09433962 | 25 | 0 | 30 |
| Hypothetical protein | 320,337–3,320,558 | 222 | 0.10291595 | 25 | 0 | 35 |
| Membrane protein insertion efficiency factor YidD | 896,338–896,544 | 207 | 0.09433962 | 26 | 0 | 30 |
| Hypothetical protein | NA | NA | 0.12864494 | 25 | 0 | 50 |
| Hypothetical protein | 895,073–894,882 | 192 | 0.09433962 | 25 | 0 | 30 |
| Hypothetical protein | 34,731–36,224 | 1,494 | 0.06346484 | 26 | 0 | 11 |
| Transcriptional regulator | 628,466–628,062 | 405 | 0.07547170 | 26 | 0 | 18 |
| YkgJ family cysteine cluster protein | 36,199–36,468 | 270 | 0.06346484 | 26 | 0 | 11 |
| Hypothetical protein | 628,964–628,698 | 267 | 0.07547170 | 26 | 0 | 18 |
| Hypothetical protein | 881,931–881,236 | 696 | 0.08919383 | 25 | 0 | 27 |
| Type 1 fimbrial protein | 151,233–151,835 | 603 | 0.04459691 | 25 | 0 | 1 |
| Hypothetical protein | 895,662–896,084 | 423 | 0.08919383 | 25 | 0 | 27 |

**Table 2 C2 unique genes.** Genes unique to C2 from pangenome analysis.

| Consensus product | Locus | Length | Percentage of pangenome | C1 | C2 | Not C1C2 |
|---|---|---|---|---|---|---|
| Protein phosphatase 2C domain-containing protein | 1,851,770–1,852,867 | 1,098 | 0.04974271 | 0 | 23 | 6 |
| OmpA family protein | 1,854,448–1,855,056 | 609 | 0.04974271 | 0 | 23 | 6 |
| OmpA family protein | 1,854,232–1,854,454 | 223 | 0.04974271 | 0 | 23 | 6 |
| Hypothetical protein | 24,841–26,628 | 1,788 | 0.07890223 | 0 | 23 | 23 |
| Transporter MotA/TolQ/ExbB proton channel domain protein | 1,852,933–1,854,228 | 1,296 | 0.04974271 | 0 | 23 | 6 |
| Kinase | 1,850,776–1,851,726 | 951 | 0.04974271 | 0 | 23 | 6 |
| Hypothetical protein | 596,554–596,781 | 228 | 0.08061750 | 0 | 23 | 24 |
| VWA domain-containing protein | 1,850,126–1,850,776 | 651 | 0.04974271 | 0 | 23 | 6 |

Four of the five genes within the boundary of the cassette have been assigned a gene name and function by bakta annotations (*Schwengers et al., 2021*). One of those gene families is the gene *tnpA*, which is crucial for IS200/IS605 family transposition (*Barabas et al., 2008*; *Morero et al., 2018*). The remaining three were potentially part of a sugar metabolic operon based on their annotations. These were (*ptsEII*)-a sugar transmembrane transporter; *malQ*, 4-α-glucanotransferase important in maltose metabolism; and *treR* the repressor of the trehalose metabolic pathway. The presence of intact *pxpB* was highly

homoplasic in *H. influenzae* (consistency Index of 0.14), which suggested a history of frequent gain and loss in the species, commensurate with cassette insertion. There was a pattern of lost and gained gene families found in other sequence types on the tree. We observed that there were 42 STs that have the exact same pattern of the lost *pxpB* gene family and concurrently gained the five gene families within the cassette. The pattern of gains and losses could be explained by multiple independent insertions of a cassette at the same location that disrupted *pxpB*. The metadata available for these 42 STs included isolates from both blood and respiratory infections. Many isolates were from infections of populations at risk of pulmonary infection such as patients with chronic obstructive pulmonary disease (COPD) and cystic fibrosis.

### C1 and C2 have unique accessory gene profiles closer to *H. influenzae* isolates from blood than sputum

Instances of NTHi infection are usually respiratory in nature, but C1 and C2 infections were found to be largely systemic. This prompted us to do a GWAS of C1 and C2 to the larger NTHi pangenome to determine whether gene C1 and C2 profiles more closely matched samples collected from sputum or blood. Out of 4,842 *H. influenzae* isolates with public genomes, 1,624 had metadata indicating the isolates were collected from a blood or system infection and 1,441 were labeled as being isolated from sputum (the other genomes did not have identifiable metadata). When this metadata is plotted on the phylogenetic tree, we do not observe notable representation of blood or sputum samples in the clade that includes both C1 and C2 samples (Fig. S5). Using Scoary, we identified 24 accessory genes that have a Bonferroni *p*-value less than 0.05 and odds ratio greater than 2 associated with blood *vs.* sputum (Table 3). Compared to the *H. influenzae* 4,842 pangenome set, representative C1 and C2 genomes were enriched in the presence of these accessory genes associated with blood infections, having 16 and 14 genes (of the 24 identified), respectively (Fig. 4).

After identifying this enrichment of potential bloodstream infection genes, we wanted to determine whether these enrichment patterns were unique to NTHi or were reflected in the larger *Haemophilus* genus. Of the 24 genes that were identified by Scoary as potentially discriminatory for bloodstream infection, only two were known virulence factors in the *Haemophilus* genus according to the VFDB. The *lsgB* gene, found in both C1 and C2 samples, is associated with *H. parasuis* virulence through its involvement in lipooligosaccharide biosynthesis sialylation (*Wang et al., 2018*). It is one of several virulence genes in *H. parasuis*, contributing to the bacterium's pathogenicity by influencing sialylated lipooligosaccharide production. Another of the 24, an *igaA1* gene, found in C1 but not C2 genomes, is a homolog of *Salmonella* membrane protein IgaA (*Tierrez & García-del Portillo, 2004*). IgA regulates bacterial regulons like RcsC-YojN-RcsB and PhoP-PhoQ, with the igaA1 allele (due to an R188H mutation) altering the expression of PhoP-PhoQ-activated (pag) genes, such as *ugd*, which is linked to lipopolysaccharide modification and colanic acid capsule synthesis (*Tierrez & García-del Portillo, 2004*).

**Table 3 Significant genes from pan-GWAS.** Table of 24 significant genes from the pan-GWAS analysis, showing gene names when applicable, the functional annotation, the Bonferroni-adjusted *p*-value from the pan-GWAS analysis, and the odds ratio from the pan-GWAS.

| Gene | Annotation | Bonferroni *p*-value | Odds ratio |
| --- | --- | --- | --- |
| xylB | xylulokinase | 3.78E−02 | 4.637255 |
| NA | Integration host factor subunit beta | 3.34E−02 | Inf |
| fbp | Class 1 fructose-bisphosphatase | 3.07E−02 | 16.225352 |
| xylA | xylose isomerase | 1.53E−02 | 6.488160 |
| NA | Hypothetical protein | 1.40E−02 | 30.208333 |
| NA | Hypothetical protein | 1.32E−02 | 4.696154 |
| dacB | Serine-type D-Ala-D-Ala carboxypeptidase | 9.87E−03 | 3.919662 |
| igaA1 | Autotransporter domain-containing protein | 2.36E−03 | 10.134375 |
| gss_2 | glutathionylspermidine synthase family protein | 2.22E−03 | 6.046512 |
| NA | Hypothetical protein | 2.08E−03 | Inf |
| NA | ABC transporter ATP-binding protein | 1.56E−03 | 4.482759 |
| ftnA_1 | Non-heme ferritin | 1.27E−03 | Inf |
| NA | YchF/TatD family DNA exonuclease | 1.05E−03 | 7.157895 |
| NA | Type II/IV secretion system protein | 4.20E−04 | Inf |
| Isg B | Lipooligosaccharide biosynthesis sialyltransferase LsgB | 4.20E−04 | Inf |
| NA | ABC transporter ATP-binding protein/permease | 3.07E−04 | 13.292929 |
| NA | Capsular polysaccharide biosynthesis protein | 1.39E−05 | 20.816327 |
| NA | ABC transporter ATP-binding protein | 1.39E−05 | 20.8163.27 |
| NA | ABC transporter permease | 1.39E−05 | 20.816327 |
| NA | Capsule biosynthesis protein | 1.39E−05 | 20.816.327 |
| NA | TolC family protein | 4.18E−06 | 9.081818 |
| sbcB | Exodeoxyribonuclease | 1.74E−07 | 23.833333 |
| NA | Na+/H+ antiporter NhaC family protein | 1.26E−08 | Inf |
| NA | Glycosyltransferase family 8 protein | 9.10E−11 | 9.868919 |

## Rates of recombination were similar in C1 and C2 to the rest of the *H. influenzae* species

Previous work in *Neisseria meningitidis* has shown that homologous recombination events can contribute to a novel pathogenicity phenotype (*Tzeng et al., 2017*). We hypothesized that allelic variation introduced by homologous recombination may have played a role in the pathogen evolution in the C1 and C2 clusters. We created a core genome alignment of 50 randomly chosen *H. influenzae* STs and the C1-1 and C2-1 genomes to predict potential regions of recombination using ClonalFrameML (*Kwong, 2017*) (Fig. 5A). As found in previous studies (*Carrera-Salinas et al., 2021*; *Gonzalez-Diaz et al., 2022*) recombination was common across *H. influenzae* genomes. There were 44 recombination events in the C1-1 genome with a rho/theta of 1.6, (rho/theta is the frequency of recombination events relative to the mutation rate). There were no recombination events identified in the C2-1 genome (rho/theta = 0.58). When we looked at the rho/theta values of the other *H. influenzae* samples we found that both C1-1 and C2-1 were within the

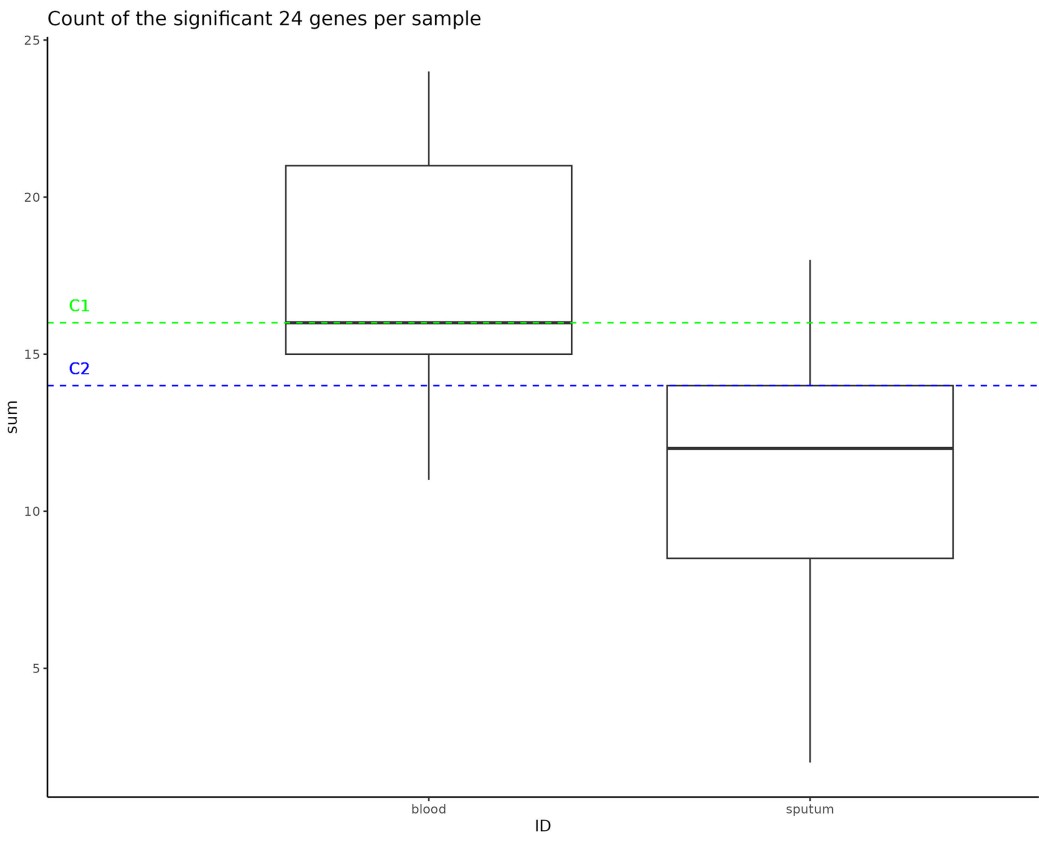

**Figure 4 Significant gene counts in all samples.** This box represents the count of the significant 24 genes found to be enriched in bloodborne samples, counted in all samples used as an input to the pan-GWAS analysis. The dotted lines represent the number of these bloodborne genes that are found in the C1 and C2 samples.

ranges of other clades (Fig. 5B), suggesting that patterns of recombination were not atypical for the species.

## DISCUSSION

Using innovative and comprehensive bacterial genotypic analytic methods, we investigated whether the emergence of two clones of NTHi (C1 and C2) associated with a novel clinical presentation of invasive disease in metropolitan Atlanta (primarily occurring among persons with well-controlled HIV) was associated with genomic changes that could have increased the virulence of clones. While we did not find genes with known functions unique to C1 and/or C2, we did identify a novel cassette encoding a polysaccharide metabolism cluster inserted in C1/C2 genomes in a manner that disrupted the *pxpB* gene. This disruption could itself be linked to a pathoadaptive phenotype. The *pxpB* gene is part of an operon including *pxpA* and *pxpC*. Single gene mutations in *B. subtilis* showed that each of the *pxpA*, *pxpB*, and *pxpC* genes were necessary and sufficient for 5-oxoproline (OP) metabolism, and deletion of any resulted in OP accumulation and slowed growth (*Niehaus et al., 2017*). Accumulation of OP causes several cellular responses in prokaryotes, including growth inhibition (*Niehaus et al., 2017*). Deletion of *pxpB* showed

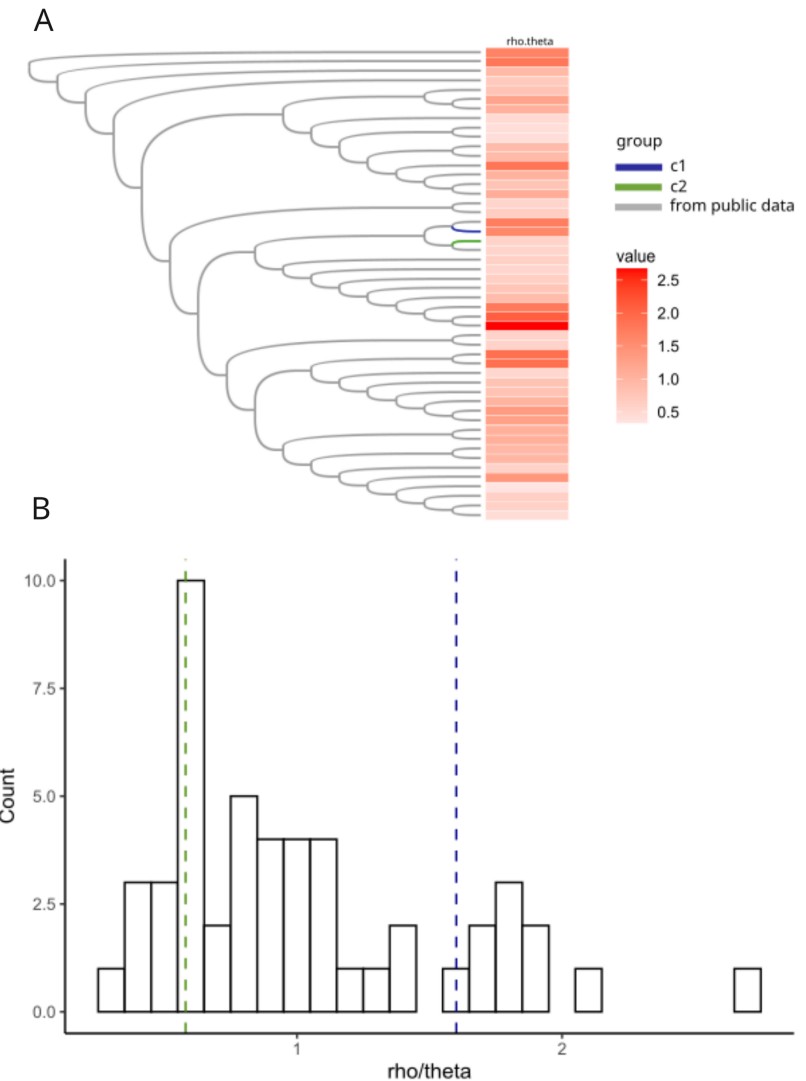

**Figure 5 Recombination analyses of C1 and C2.** Recombination results from ClonalFrameML. (A) Neighbor-joining tree of a subset of 50 randomly selected sequence-type representative genomes of *H. influenzae* plus C1 and C2. (B) Histogram of distribution of rho.theta values across the dataset. Two vertical dashed lines indicate specific rho.theta values: 1.60115 (blue) corresponding to C1 and 0.579753 (green) corresponding to C2.               

aberrant DNA recombination within a large genetic interaction screen in *E. coli*\xEF\xB7 \x9F (*Kumar et al., 2016*). The disruption of *pxpB* should therefore be associated with a fitness deficit, so it is interesting that it is the target for disruption in several *H. influenzae* lineages. This might suggest a possible tradeoff for a pathoadaptive trait and should be investigated further. should therefore be associated with a fitness deficit, so it is interesting that it is the target for disruption in several *H. influenzae* lineages. This might suggest a possible tradeoff for a pathoadaptive trait and should be investigated further.

C1 and C2 possess a suite of virulence genes associated with bloodstream infections, only two of which are associated with other *Haemophilus* species (*IsB, igaA1*). However, three of the NTHi virulence genes (*dacB*, *fbp*, and *sbcB*) that are associated with blood

infections have been associated with virulence mechanisms for other pathogens. Interestingly, *dacB* and *fbp* are the only genes also found core genes in our pangenome analysis. The *dacB* gene encodes for a serine-type D-Ala-D-Ala carboxypeptidase, and it appears to have the potential to influence virulence in the context of peptidoglycan (*Lerner et al., 2012*; *Abdullah et al., 2014*). Studies have shown that disabling this gene, as well as its counterpart *dacA*, in pneumococci led to significant attenuation of the bacteria in infected mice (*Abdullah et al., 2014*). Additionally, mutants lacking *dacB* and *dacA* exhibited enhanced uptake by professional phagocytes and decreased adherence to lung epithelial cells. In another context, a mutation in dacB was associated with changes in peptidoglycan structure, including the release of different peptidoglycan fragments, highlighting its role in peptidoglycan metabolism (*Lerner et al., 2012*). The *fbp* gene, which encodes fructose-1,6-bisphosphatase, exhibits varying effects on virulence in different bacterial and protozoan species. In *Brucella*, the loss of *fbp* does not impact virulence, suggesting that it is not essential for full virulence in laboratory models (*Zúñiga-Ripa et al., 2018*). Similarly, in *Brucella suis* biovar 5, *fbp* is not required for full virulence in laboratory models (*Zúñiga-Ripa et al., 2018*). However, in *Leishmania*, the gluconeogenic enzyme fructose-1,6-bisphosphatase encoded by fbp is essential for virulence, as mutants lacking this enzyme can persist in mice but fail to generate normal lesions (*Naderer et al., 2006*). This suggests that *Leishmania* relies on fructose-1,6-bisphosphatase for virulence, possibly due to its dependence on non-glucose carbon sources in glucose-poor phagosomes. Additionally, *fbp* has been identified in a screen in *Staphylococcus*, but its specific role in virulence in this context is not detailed in the provided information (*Srinivasan et al., 2022*). The *sbcB* gene, which encodes exodeoxyribonuclease I, is a recognized component of virulence in *Salmonella* (*Cano et al., 2002*). Research has shown that mutants of *Salmonella* lacking the RecBC function, in which *sbcB* is involved, are avirulent in mice and incapable of growing inside macrophage (*Cano et al., 2002*). This finding highlights the critical role of the RecBCD recombination pathway, in which *sbcB* plays a part, in Salmonella's virulence. This pathway is essential for repairing double-strand breaks generated during DNA replication and is proposed to be necessary for systemic infection by *S. enterica*, as it likely facilitates DNA replication within phagocytes during infection, notably the other pathogens included don't necessarily cause joint manifestations.

The comparative genomic approach described here is limited to events involving gain and loss or recombination-driven allelic change in genes with known virulence functions; within those limitations, our data do not reveal unusual patterns of virulence genes in C1/ C2. There are genomic changes that could cause hypervirulence but would not be detected using the methodology implemented in this analysis, such as rare SNPs, particularly in regulatory genes, genomic rearrangements, and the gain of virulence genes of unknown function. One future exploratory approach may include evaluating potential differences in gene expression in virulence models between C1/ C2 and other *H. influenzae* in conditions that simulate bloodstream infection.

## CONCLUSIONS

Our comparative genomic analysis of emerging non-typeable *Haemophilus influenzae* (NTHi) clones C1 and C2, associated with invasive infections in Atlanta, has provided several key insights into their genetic characteristics and potential virulence factors. While we identified some genetic markers potentially linked to systemic infections, our study did not reveal definitive unique genetic factors that distinguish these clones as more virulent than other *H. influenzae* strains.

Our investigation revealed high genetic similarity within C1 and C2 clusters, with limited SNP diversity in their core genomes. Notably, we did not identify any unique accessory genes in C1 and C2 that could definitively explain increased virulence or invasiveness. However, we observed a consistent loss of the *pxpB* gene in both C1 and C2 isolates, replaced by a novel mobile cassette potentially involved in sugar metabolism. This finding suggests a possible pathoadaptive change, though its exact implications require further study.

Interestingly, C1 and C2 isolates showed enrichment of accessory genes associated with bloodstream infections, which may contribute to their invasive nature. The recombination rates in C1 and C2 were like other *H. influenzae* clades, indicating that unusual recombination events are unlikely to explain their emergent behavior. Unexpectedly, we also observed deletions in known virulence genes, suggesting possible attenuation rather than enhancement of virulence.

The expansion of these clones in a vulnerable population may reflect both chance introduction and potential adaptations to the host environment. Our data suggest that the emergence of these invasive NTHi strains likely results from a complex interplay of bacterial genetics, host factors, and epidemiological circumstances.

Further research is needed to fully understand the implications of our findings. Future studies should investigate the functional implications of the *pxpB* gene loss and the acquired mobile cassette. It will be crucial to explore host factors potentially contributing to the invasiveness of these strains, as well as examine non-genetic factors influencing their spread. Additionally, analyzing gene expression patterns under conditions simulating bloodstream infections may provide valuable insights into the virulence mechanisms of these strains.

While our genomic analysis has not provided a clear genetic explanation for the increased invasiveness of C1 and C2 clones, it has highlighted several areas for further investigation. These findings underscore the importance of continued surveillance and research to understand and manage the evolving landscape of invasive NTHi infections. The implications of these genetic findings on clinical management and prevention strategies for invasive NTHi infections require further elucidation. Our study contributes to the growing body of knowledge on NTHi genomics and sets the stage for more targeted investigations into the factors driving the emergence of invasive strains in vulnerable populations.

### Funding

Brianna J Bixler and Timothy D. Read were supported by the Office of Advanced Molecular Detection, Centers for Disease Control and Prevention Cooperative Agreement Number CK22-2204 through contract 40500-050-23234506 from the Georgia Department of Public Health. Brianna J Bixler was also supported by NIH F31ES031845. Lauren F. Collins is also supported by the National Institute on Aging (NIA) of the NIH (award number K23AG084415). The Emerging Infections Program (EIP) project described was supported by Grant ID NU50CK000645 from the Centers for Disease Control and Prevention (CDC). Its contents are solely the responsibility of the authors and do not necessarily represent the official views of the CDC. The funders had no role in study design, data collection and analysis, decision to publish, or preparation of the manuscript.

### Grant Disclosures

The following grant information was disclosed by the authors:
Office of Advanced Molecular Detection, Centers for Disease Control and Prevention Cooperative Agreement: CK22-2204.
Georgia Department of Public Health: 40500-050- 23234506.
National Institute on Aging (NIA) of the NIH: F31ES031845 and K23AG084415.

### Competing Interests

Timothy D. Read is an Academic Editor for PeerJ.

### Author Contributions

- Brianna J. Bixler conceived and designed the experiments, performed the experiments, analyzed the data, prepared figures and/or tables, authored or reviewed drafts of the article, and approved the final draft.
- Charlotte J. Royer conceived and designed the experiments, performed the experiments, analyzed the data, prepared figures and/or tables, authored or reviewed drafts of the article, and approved the final draft.
- Robert A. Petit III conceived and designed the experiments, performed the experiments, analyzed the data, prepared figures and/or tables, authored or reviewed drafts of the article, and approved the final draft.
- Abraham G. Moller performed the experiments, analyzed the data, authored or reviewed drafts of the article, and approved the final draft.
- Samantha Sefton conceived and designed the experiments, performed the experiments, analyzed the data, authored or reviewed drafts of the article, and approved the final draft.
- Stepy Thomas conceived and designed the experiments, performed the experiments, analyzed the data, authored or reviewed drafts of the article, and approved the final draft.
- Amy Tunali performed the experiments, analyzed the data, authored or reviewed drafts of the article, and approved the final draft.

- Lauren F. Collins performed the experiments, analyzed the data, authored or reviewed drafts of the article, and approved the final draft.
- Monica M. Farley performed the experiments, analyzed the data, authored or reviewed drafts of the article, and approved the final draft.
- Sarah W. Satola conceived and designed the experiments, performed the experiments, analyzed the data, authored or reviewed drafts of the article, and approved the final draft.
- Timothy D. Read conceived and designed the experiments, performed the experiments, analyzed the data, prepared figures and/or tables, authored or reviewed drafts of the article, and approved the final draft.

## DNA Deposition

The following information was supplied regarding the deposition of DNA sequences:

The sequences described in this study are available at NCBI BioProject: PRJNA544724.

## Data Availability

The data, codes, and commands are available at Zenodo:

- Robert A. Petit III. (2025). royercj/gaeip-nthi: Bixler-2024-gaeip-nthi (bixler-2024). Zenodo. https://doi.org/10.5281/zenodo.14861587.

- briannajeanne. (2025). royercj/NTHi: Bixler-2024-NTHi (bixler-2024). Zenodo. https://doi.org/10.5281/zenodo.14861572.

## Supplemental Information

Supplemental information for this article can be found online at http://dx.doi.org/10.7717/peerj.19081#supplemental-information.

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
