# Peer review of "Comparative genomic analysis of emerging non-typeable Haemophilus influenzae (NTHi) causing emerging septic arthritis in Atlanta"

_PeerJ, doi:10.7717/peerj.19081_

## Round 0.1 · original submission · Major Revisions

While both reviewers agreed that this was a well-written manuscript, they have identified a number of areas that require some clarification and revision. While none of these on their own should prove onerous, due to the excellent level of detail and number of comments provided by the reviewers, I have selected 'Major revisions' to provide you with a little more time to address these.

·

Basic reporting

The authors have used professional English that are understandable for international audience. There was minimal typing or grammatical error. However, major improvement is needed, specifically in the following areas, sorted from the most to least important:
1. Conform to professional article structure; figure and table labelling; and availability of correct, and complete raw data of all informations relevant in the study.
2. Self-containment to stated hypotheses.
3. Literature references in the Introduction section.

These are the specific feedback in the area:
General - Background: The authors have sufficiently shown the context of the research and why it is important. However, only one original research article was cited mostly throughout the background (reference number 5). Even though reference no. 5 was the source of the H. influenzae isolates investigated in the current research, the authors should add additional relevant references on what has been known about NTHi causing invasive diseases.
Line 52-53: Please specify the name and function of the 2 genes flanking the IS1016 found in C1 isolates. Additionally, if these 2 genes are found in all C1 isolates, this should be indicated as well.
Line 83: Please correct a typing error after the Victors database link. Please double check whether this is the correct link because at the time of the review the web page showed "The requested URL was not found on this server." when the link was clicked
Line 83-84: Sheet 1 on supplementary table S1 only shows a short list of gene families without details on their protein products etc. Please make sure the appropriate data of curated Virulence Factors have been included in the Supplementary Table attached for manuscript submission.
Line 116: Need to deposit this python script in a public repository and cite the link to the repository.
Line 119-121: This sentence needs major improvement for better clarity:
- Does ""For the remaining sequence types.."" at the start of the sentence imply that the Python script will automatically pass Georgia isolates with ST 164 and ST 1714, but not the rest?
- Samples rank ""gold, silver, bronze"": what does each rank mean and what are the parameters used to decide the rank? Additionally, what is the threshold to include/exclude isolates, i.e. are only ""gold"" isolates included?"
Line 114-135: The paragraph of Line 114 - 135 is in general confusing and will benefit from a supplementary figure of a step-by-step workflow for including/excluding isolates.
Line 142: Abbreviation (rpkm) needs to be stated in detail what it stands for.
Line 160-162: Depositing a list of isolates in a data repository is OK but I highly suggest to provide the list of all isolates included from NCBI SRA with their metadata and important results from the genomic analyses, as a supplementary table within the article.
Line 195: The following notation in brackets doesn't fit: ""1,107 intermediate accessory (95%< x <= 5%) and 4,085 rare accessory (x > 5%)"".
Should it be ""1,107 intermediate accessory (5%< x < 95%) and 4,085 rare accessory (x <= 5%)""?"
Figure 3: In text pxpB was mentioned as gene family of interest; it should be make clear which column of metadata block represents this gene (i.e. provide a gene name as the column label)
Line 206: "Table 2" should not appear before "Table 1". Table naming should be in order.
Table 1, Table 2, Table 3: In the PDF file, these tables had subtitles and each one of them started with ""Table 1:"". This should be corrected.
Gene family naming from PIRATE doesn't mean anything; if the experiment was repeated using an identical set of genome, the naming will change for the same coding sequence. It is more useful to use NCBI gene symbol notation (i.e. pxpB, atpG) or even better, an NCBI gene accession ID from its H. infulenzae reference genome."
Line 210: what are these 583 samples?
Line 209 - 215: This paragraph needs major improvement. It is unclear what is described.
Line 219 - 222: Do these sentences correspond to Figure 3? If yes, figure 3 needs to be cited
Line 254 - 255: Figure s5 shows "rho.theta" on the metadata block - how is it associated with "representation of blood or sputum samples" in the clade? From my understanding, rho/theta is a recombination statistics from ClonalFrame ML.
Line 280 - 297: This is just a reiteration of the findings. Avoid this in Discussion.
Line 312 - 314: Refer to the comment on Line 228 - 231
Line 314 - 315: There was no mention of this in the result section on important virulence factors detected in C1 and C2 samples. It should be mentioned in the result section, and then the reason why it is an important finding can be elaborated in the Discussion section, in relation to existing literatures.
Line 324 - 351: This part of the discussion also elaborates the results of the pan-GWAS analysis. If this is what the authors wanted to highlight in their study, then the narrative of the article should be changed and focused on this throughout the article. It is understandable that the authors explored accessory genes associated with bloodstream infection, and evaluated whether these genes were more frequent in the C1 C2 NTHi Atlanta isolates collection. However, this was not apparent in the current structure of the article.

Experimental design

The submitted article is in line with PeerJ Aims and Scope. The authors have a well-defined aim and research question with clear understanding of what is the knowledge gap and how to answer that. The authors had shown efforts in providing details in the Method section; however, there were important experiments that are not described in this section or described but not sufficient for other people to replicate.

These are the specific feedback in the area:
General - Background: The authors stated a clear research question and how answering it can fill the knowledge gap on invasive NTHi genomics.
Line 71: Please elaborate what C1-1 and C2-1 isolates are and how they are chosen since this has not been mentioned. The sentence in the Result section Line 167-169 should be moved to this section to improve clarity.
Line 102-104: This paragraph describes the process of obtaining publicly available genomes; however, this sentence mentions "reference genes" so it doesn't fit here. The "105 reference genes" from reference no 11 were virulence factors as subtracted from the Virulence Factors Database. Therefore this should be consolidated in the "H. influenzae virulence protein database" section.
Line 118, Line 129, Line 133: The term ""representative set"" and ""representative samples"" are not clearly defined:
- Are they the same?
- Do they contain all isolates from Georgia (despite the ST and whether they belong to C1, C2, or neither) AND the global dataset extracted from NCBI SRA?
This unclarity will make it impossible for method replication in the future.
Line 134-135: Please add the method of phylogenetic analysis (i.e. Maximum-Likelihood, Neighbor-Joining, etc).
If core genome alignment was used to reconstruct the phylogenetic tree, any particular reason why the core genome MLST scheme available from PubMLST was not utilised to produce the core genome alignment?
Line 147: If STs were chosen randomly, were ST 164 and ST 1714 (the two main STs in this study) were accounted for in the recombination analysis? From Figure 5 it can be concluded that C1 and C2 isolates were included in this analysis. However, this has to be stated explicitly in the method.
Line 129: Were capsulated H. influenzae isolates from NCBI SRA excluded from the "representative set"? If yes, please specify the method and tool used for defining capsulated strains. If not, please provide a justification for why they were included as this study focused on invasive NTHi strains
Line 228 - 231: There was no mention of comparing the loss and gain of gene families to other Haemophilus species in the method section. What is the purpose if this analysis? This should be stated in the method section.
Line 255 - 263: There was no mention of conducting a GWAS tool in the method section. Additionally, this experiment is not in line with the aim of the study. It was stated that the study aimed to performed comparative genomic analyses of C1 C2 NTHi isolates from Atlanta in the context of global H. influenzae strains. Previously it was also stated that in the cluster where C1 and C1 isolates both belonged, there was no blood or sputum samples. Why is it relevant to conduct GWAS comparing blood vs sputum?

Validity of the findings

This study is an important extension of previous study and the rationale and benefit to the literature is clearly stated. However, there was a lack of completeness in providing all utilised data (this was mostly mentioned in the "Basic reporting" part above). Additionally, some improvement in the Conclusions section is encouraged.

Below is the specific feedback in the area:
Line 140: Does the ""completed reference genome of each"" refer to C1-1 and C2-1 isolates which were resequenced using Nanopore and Illumina to form a complete genome?
If yes, this raise the following question for identifying deletion regions within C1 and C2. The C1-1 and C2-1 isolate was chosen randomly. When a deletion region was detected, how can the authors be sure that it was a true deletion region among isolates, instead of the reference genome acquiring genes in the same region?
If not, please specify in the method section what is the reference genome used and the NCBI genome accession."
Line 177, Line 184: It is important to add in the text (at the start of the paragraph) that "deletion regions" were defined as "unique regions of low coverage found in at least 1 sample", as stated in the Figure 2 legend.
Line 190-191: In the method section (Line 114 - 135), the selection process of publicly available H. influenzae genomes was multilayered, indicating some genomes did not pass the quality check step, etc. However, this line indicates that all genomes (N = 4,842) were compared to C1 and C2 in the end. My suggestion for the method section Line 114-135 was to make a workflow diagram - this could be supplemented with the number of genomes passing the quality check at each stage. Additionally, the genomes, their corresponding SRA accession ID, and relevant metadata should be provided as supplementary file.
Line 209-210: Please make sure the appropriate data of curated Virulence Factors have been included in the Supplementary Table attached for manuscript submission and cited in this sentence.
General - Conclusions: The conclusion should be made clear and succinct (1 - 2 paragraphs) stating what is the main finding of the study, its relevance, and what should be done in the future. Avoid reiteration of Results or Discussion.

Reviewer 2 ·

Basic reporting

Well written manuscript easily understandable and amazingly readable given the technical issues that are involved in the processes studied/analyzed/reported. Meets all standards of basic reporting. My only recommendation here is that some of the information about the clinical population studied and reported in original 2019 JAMA article be summarized in a single paragraph in the Introduction section. That would include the reported numbers of cases and epidemiologic data. In this way, the extent of the remainder of the paper can be better understood including the number of H. flu isolated and studied in this study.

Experimental design

Well explained experimental design and written in understandable English.

Validity of the findings

Excepti0onally valid information assuming that all procedures were followed to the exact protocol of processing. The interpretation of this is very interesting and the implications of potentially transmissible genes is astonishing.

Additional comments

No additional comments.

---

## Round 0.2 · accepted · Accept

The authors have addressed all of the reviewers' comments. The previous reviewer was invited, but did not feel the need to re-review; I have assessed this revision myself and feel that the manuscript is now ready for publication.